# Host-Directed Therapies and Anti-Virulence Compounds to Address Anti-Microbial Resistant Tuberculosis Infection

**Raphael Gries [1,2], Claudia Sala [3] and Jan Rybniker [1,2,4,*]**

1   Department I of Internal Medicine, Division of Infectious Diseases, University of Cologne, 50931 Cologne, Germany; raphael.gries@uk-koeln.de
2   Center for Molecular Medicine Cologne, University of Cologne, 50931 Cologne, Germany
3   Fondazione Toscana Life Sciences, 53100 Siena, Italy; c.sala@toscanalifesciences.org
4   German Center for Infection Research (DZIF), Partner Site Bonn-Cologne, 50931 Cologne, Germany
*   Correspondence: jan.rybniker@uk-koeln.de; Tel.: +49-221-478-89611; Fax: +49-221-478-5915

**Abstract:** Despite global efforts to contain tuberculosis (TB), the disease remains a leading cause of morbidity and mortality worldwide, further exacerbated by the increased resistance to antibiotics displayed by the tubercle bacillus *Mycobacterium tuberculosis*. In order to treat drug-resistant TB, alternative or complementary approaches to standard anti-TB regimens are being explored. An area of active research is represented by host-directed therapies which aim to modulate the host immune response by mitigating inflammation and by promoting the antimicrobial activity of immune cells. Additionally, compounds that reduce the virulence of *M. tuberculosis*, for instance by targeting the major virulence factor ESX-1, are being given increased attention by the TB research community. This review article summarizes the current state of the art in the development of these emerging therapies against TB.

**Keywords:** tuberculosis; *Mycobacterium tuberculosis*; host-directed therapy; anti-virulence compounds

---

## 1. Introduction

*Mycobacterium tuberculosis*, the etiological agent of human tuberculosis (TB), is thought to latently infect approximately one fourth of the world's population and is responsible for over one million deaths every year [1], thus representing the leading cause of mortality by an infectious disease worldwide. Immunodeficiency caused by HIV [2] and co-morbidities like diabetes [3] constitute additional risk factors for the development of active TB disease.

The current anti-TB therapy consists of a combination of four antibiotics (rifampicin, isoniazid, pyrazinamide and ethambutol) that must be administered for at least 6 months in case of drug-sensitive pulmonary TB infection [4]. However, *M. tuberculosis* displays increased resistance to first-line drugs, which has resulted in multidrug-resistant (MDR) and extensively drug-resistant (XDR) TB cases [5]. Second-line treatment regimens are therefore employed but require longer duration to be effective and are associated with severe side effects that frequently decrease patient compliance [6].

To address the increasing need for new and potent therapeutic options against TB, alternative approaches are being explored. These include host-directed therapy (HDT) and anti-virulence compounds. Within the first choice, a number of molecules that reduce inflammation, modulate autophagy and potentiate the immune response are currently in preclinical and in clinical trials. On the other hand, drugs that affect *M. tuberculosis* ability to infect and kill host cells represent a promising complement to standard antibiotic treatment.

Here we review the current state of research in the areas of HDT and anti-virulence drugs as complementary approaches to TB therapy.

## 2. Host Directed Therapy

### 2.1. Promoting Phagosome Maturation and Enhancing Autophagy

Autophagy is a natural process which protects cells against unfolded proteins and potentially dangerous aggregates, viral and bacterial infections. By means of autophagy, macrophages deliver toxic macromolecules, organelles and phagocytosed pathogens to lysosomes for degradation [7]. Modulating autophagy in order to promote bacterial killing represents one example of HDT against TB infection. This statement is supported by studies conducted by Gutierrez and co-workers who reported that stimulation of autophagy in macrophages promotes phago-lysosome maturation and impacts mycobacterial survival [8]. More recent investigations revealed increased susceptibility to TB in mice defective in autophagy pathways [9] and a positive interplay between autophagy and interferon-gamma (IFN-$\gamma$) in TB patients [10].

#### 2.1.1. mTOR Inhibition

The best described autophagy inducer is rapamycin (sirolimus), used in patients who underwent organ transplantation [11]. Rapamycin, a macrolide produced by *Streptomyces hygroscopicus* and originally discovered on Easter Island (Rapa Nui for the inhabitants, hence the name given to the compound), inhibits TOR, the Target Of Rapamycin [12]. The mammalian Target of Rapamycin mTOR is a negative regulator of autophagy [13]. Its clinical use in infectious diseases is restricted due to its broadly immunosuppressive effects. In addition, rapamycin is metabolized by CYP3A4 [14], a hepatic enzyme induced by the first-line TB drug rifampicin, thus hampering its exploitation as HDT in TB patients. However, other molecules capable of inducing autophagy have been discovered and are now under development. Among these, vadimezan [15], Tat-beclin 1 fusion peptide [16], the calcium-channel blocker verapamil [17] and the rapamycin analogue everolimus [18]. In particular, verapamil was shown to be efficacious when combined with rifampicin and with the recently approved medication bedaquiline in mouse models of infection, where it increased the bioavailability of the antibiotic [19–21]. On the other hand, everolimus, an anti-cancer agent, may be repurposed as an anti-TB HDT therapeutic capable of inhibiting mTOR although, as with rapamycin, immunosuppression [22] and toxicity [23] may represent issues in the clinical development as an anti-TB drug.

#### 2.1.2. Metformin

One of the most promising drugs that promotes autophagy is currently used for treatment of type 2 diabetes: metformin [24]. This compound is characterized by a good safety profile, activates 5'-adenosine monophosphate-activated protein kinase (AMPK), induces production of mitochondrial reactive oxygen species (mROS), which are deleterious to *M. tuberculosis* and was found to reduce the severity of TB disease in humans [25]. Despite these promising data, combination therapies which involved metformin in mice had contradictory results. While in one case metformin enhanced the activity of isoniazid and ethionamide [25], in another one it did not improve the efficacy of the combined first-line drugs [26]. Recent retrospective studies reported a protective effect for metformin against reactivation of latent TB in diabetic patients [27–29]. Phase II clinical trials have been initiated (Table 1).

Activity of the major virulence factor ESX-1 can be blocked by compounds BBH7 and BTP15. While BBH7 hinders the secretion mechanism by inducing zinc stress, BTP15 was shown to act by downregulating expression of the *espA-espC-espD* operon upon interaction with MprB. The secreted proteins MptpB, SapM and Zmp1, which prevent phagosomal maturation, can be directly targeted extracellularly by their respective inhibitors. The dedicated secretion systems for these three proteins

are not described yet. Gene expression of PhoP-dependent virulence genes can be controlled by PhoP inhibitors which prevent binding of PhoP to specific promoter regions, thus affecting transcription.

### 2.1.3. Imatinib and Other Tyrosine Kinase Inhibitors

A key feature of *M. tuberculosis* is represented by its ability to inhibit phago-lysosome fusion, and thus potentially limit the efficacy of the autophagy process, thanks to the presence of specific virulence factors like lipoarabinomannan in the cell wall [30], the ESX-1 secretion system [31,32] and other key components such as the Eis protein which modulates autophagy and inflammation and suppresses host innate immune responses [33]. This inhibitory effect can be overcome by tyrosine kinase inhibitors, such as imatinib and the second-generation inhibitors nilotinib and dasatinib, which target the BCR-ABL fusion protein and are used for treating chronic myeloid leukemia [34]. Different studies explored the effect of imatinib on *M. tuberculosis*-infected macrophages and revealed that it increases acidification of lysosomes thereby halting bacterial multiplication [35]. Moreover, imatinib was shown to reduce the number of granulomatous lesions in mice and to act synergistically with first-line anti-TB drug rifampicin [36].

### 2.1.4. Statins

Statins, i.e., agents that lower cholesterol through inhibition of the biosynthetic pathway, also impact autophagy [37]. Given the relevance of cholesterol in *M. tuberculosis* persistence [38], statins have received considerable attention. Indeed, in addition to their cholesterol-lowering effect, statins decrease lipid body biogenesis and limit *M. tuberculosis* survival [39]. Additionally, it was discovered that atorvarstatin potentiates the effect of rifampin in *M. leprae* infection of the mouse footpad [40]. These substances are currently investigated in clinical trials (Table 1).

### 2.2. Vitamin D and the Induction of Anti-Microbial Peptides

Anti-microbial peptides like cathelicidins are components of the innate immune system whose synthesis is induced by mycobacterial ligands through binding to Toll-like receptors (TLRs), especially TLR2 and TLR9 [41]. Cathelicidin LL37 represents a major example of this class of molecules, is expressed by neutrophils and macrophages and participates in anti-TB defense through pore-forming capability in the bacterial membranes [42,43].

It has been shown that vitamin D promotes synthesis and release of LL37 [44], which in turn helps in autophagy [45,46]. Moreover, vitamin D enhances the ability of monocytes to respond to interferon gamma (IFN-γ) [47]. Various clinical trials which included vitamin D in addition to the standard regimen have been performed, sometimes with variable results [48–50]. It seems evident that key issues for successful use of vitamin D in TB therapy are proper dosing and possibly also genetic background and comorbidities of the patient. A recently published study by Aibana and co-workers suggested that vitamin D deficiency is associated with increased probability of developing TB in HIV-positive people [51]. However, further investigations are needed to clarify whether vitamin D supplementation might play a significant role in reducing the risk of TB.

Another vitamin whose antitubercular effects have been evaluated is vitamin A, which limits *M. tuberculosis* replication in macrophages by promoting acidification [52,53]. However, while studies in rats showed a beneficial impact of vitamin A supplementation [54], the same was not observed in humans [55,56].

Regulation of anti-microbial peptide expression is also controlled by histone deacetylase inhibitors (for instance 4-phenylbutyrate) through epigenetic mechanisms [57,58]. In the context of *M. tuberculosis* infection of human macrophages, it was demonstrated that phenylbutyrate, alone or in combination with vitamin D3, was able to counteract the suppressive effect of the bacilli on LL-37 expression, thus promoting autophagy [59].

In addition to cathelicidins, another group of anti-microbial peptides plays an important role in anti-TB mechanisms. These are defensins. Defensins are arginine-rich, cationic peptides resistant to

proteolysis. They are usually stored in the granules and in the lysosomes of innate immune cells, such as neutrophils, and are released upon pathogen invasion [60]. *M. tuberculosis* stimulates production of beta defensin-2 (HBD-2), which reduces bacterial multiplication and has a chemotactic effect [61]. Despite these features, exploitation of HBD-2 in HDT against TB is far from clinical use, due to high costs and poor stability in vivo [62]. Clinical trials where these compounds are being investigated are listed in Table 1.

### 2.3. IFN-γ and IL-2 as Adjunct Therapy

Production of anti-microbial peptides and other antimicrobial activities exerted by macrophages are stimulated by a panel of cytokines that include TNF (Tumor Necrosis Factor), IFN-γ and interleukin 1 (IL-1). While IFN-γ plays its major role in promoting autophagy and phagosome maturation, TNF increases IFN-γ responsiveness and IL-1 counteracts the detrimental effects of Type I IFN in TB [63]. HDT against TB infection includes IFN-γ and modulators of TNF, which will be discussed later in this review. Concerning IFN-γ, it was demonstrated that its administration to TB patients via the aerosol route is well-tolerated and reduces time to sputum conversion while improving lung repair after the disease [64–66]. However, the role of IFN-γ in controlling TB is still under debate, as reported in a study by Sakai and co-workers, who showed that contribution of CD4-T cell derived IFN-γ is limited and, even worse, sometimes detrimental [67]. Another clinical study, where IL-2 was added during the first month of anti-TB treatment resulted in no benefit [68], thus questioning the relevance of adding cytokines to the existing therapy. Possible side effects and treatment costs should also be considered when exploring the administration of cytokines to TB patients.

### 2.4. Inhibition of M. tuberculosis Induced Inflammation and Host Cell Death

#### 2.4.1. The Role of Corticosteroids in TB Treatment

It sounds counterintuitive to address the problem of active TB with anti-inflammatory drugs. However, for some clinical manifestations of the diseases, reduction of inflammation by using adjunctive corticosteroids has already become a well-established and lifesaving treatment approach. Addition of dexamethasone or prednisolone, two potent corticosteroids, to the antibiotic regimen for treatment of TB meningitis improves survival and is considered as a valid therapeutic approach for TB affecting the central nervous system (CNS) [69], although care should be taken since individual responses to steroid treatment might differ. Several studies have tried to improve the outcome of pulmonary TB by lowering the inflammatory response using high doses of corticosteroids in combination with antibiotics. While it was found that this therapy leads to faster resolution of symptoms and lesions in radiographic examinations and a more rapid discharge from hospitals, a statistically significant survival benefit could not be shown (Table 1) [70–72]. In addition, high dose corticosteroids may result in serious side effects such as diabetes and psychiatric symptoms. Today, corticosteroids remain the treatment of choice in specific clinical situations such as CNS TB or hyperinflammatory syndromes e.g., the immune reconstitution inflammatory syndrome (IRIS) in HIV/TB co-infected patients. Investigations at the molecular level proved that dysregulation of inflammasome signaling and of secretion of various cytokines, including IL-1γ, was associated with TB-IRIS in patients infected by HIV [73,74], thus supporting the inclusion of corticosteroids in the treatment of TB patients at risk of developing IRIS [75]. Despite these evidences, a broader application of the drugs in TB treatment is currently not justified. However, clinical studies as well as ex vivo and in vivo experiments performed with these substances indicate that a more specific or tailored modification of the TB inflammatory response may provide a suitable approach to improve patient outcomes. Understanding the exact mechanism of action of corticosteroids in TB may help overcome this hurdle. Corticosteroids are broadly immunosuppressive drugs with multiple modulatory effects on leukocytes once bound to the main target, the corticosteroid receptor. Downstream effects include repression of pro-inflammatory transcriptional regulators like NF-κB as well as impaired release of cytokines such as TNFα and IL-1 [76]. In addition, corticosteroids

such as dexamethasone seem to have an *M. tuberculosis* specific inhibitory effect on necrotic host cell death in vitro [77]. This effect seems to depend on inhibition of p38 MAP kinase which impairs mitochondrial membrane stability upon infection with *M. tuberculosis*. p38 MAP kinase is activated during *M. tuberculosis* infection in vitro and in vivo and represents a possible host directed target with several clinically tested small molecule inhibitors available for repurposing.

### 2.4.2. Non-Steroidal Anti-Inflammatory Drugs (NSAID) and Leukotriene Inhibitors

A series of mouse studies have shown beneficial effects of non-steroidal anti-inflammatory drugs (NSAIDS) such as aspirin, diclofenac and ibuprofen when used alone or in combination with common antibiotics in *M. tuberculosis*-infected mice. The main mechanism of action seems to be inhibition of prostaglandin synthesis via inhibition of cyclooxygenase 1 and 2. Prostaglandins are known drivers of tissue damaging inflammation. It is important to note that diclofenac was shown to possess growth inhibitory effects on the bacterium itself in addition to its anti-inflammatory properties. The substances have been extensively discussed elsewhere [78]. NSAID Clinical trials initiated recently are listed in Table 1. Another category of anti-inflammatory drugs is represented by leukotriene receptor antagonists, such as zafirlukast, which was reported to have anti-mycobacterial activity in vitro and cause alterations in the transcription profile in *M. tuberculosis* [79]. The potential of these drugs in HDT against TB deserves deeper investigation given the role for leukotriene A(4) hydrolase (LTA4H) demonstrated by Tobin and colleagues in animal models of infection [80,81].

### 2.4.3. Necrosis

Necrotic host cell death is a highly dynamic research field increasingly linked to the release of pro-inflammatory cytokines. A better understanding of the mechanisms of *M. tuberculosis* induced cell death may provide additional starting points for HDTs. Most studies have been focusing on cell death in macrophages, however, necrosis of other cell types such as neutrophils seems to play a pivotal and additive role in *M. tuberculosis* pathogenicity. *M. tuberculosis* released by necrotic neutrophils displays improved survival and growth once phagocytosed by adjacent macrophages [82]. Neutrophil necrotic cell death is driven by reactive oxygen species (ROS) which can be abrogated by ROS inhibitors. In addition, ROS and nitric oxide (NO) have been found to show antimicrobial activity and to modulate neutrophil recruitment to the granuloma [83]. While ROS seems to increase cytokine production and to inhibit inflammasome activation, NO shows a regulatory effect on macrophages with increased expression of hypoxia-inducible factor 1 alpha (HIF-1$\alpha$) and repression of nuclear factor kappa-light-chain-enhancer of activated B cells (NF-$\kappa$B) [84,85]. A recent study highlights a role for ferroptotic cell death in TB. Ferroptosis is a type of regulated necrosis induced by accumulation of free iron and toxic lipid peroxides which seems to be mediated by decreased levels of glutathione peroxidase-4 (Gpx4) upon *M. tuberculosis* infection in vitro. Intraperitoneal treatment of *M. tuberculosis* infected mice with ferrostatin, a ferroptosis inhibitor resulted in reduced lung pathology and decreased bacterial load [86]. In addition to ferroptosis, efferocytosis (the physiological process of removing apoptotic cells by macrophages) is an anti-bacterial mechanism that seems to play a relevant role in TB as well [87]. Indeed, efferocytosis of apoptotic neutrophils was shown to improve control of *M. tuberculosis* in an in vitro model of HIV-*M. tuberculosis* macrophage co-infection [88,89].

### 2.4.4. TNF and TNF-Mediated Signaling

Further downstream of intracellular mediators or regulators of cell death, inflammation and cytokine release, there are more direct targets amenable to therapeutic interventions. These include the cytokines themselves. Biologicals targeting TNF$\alpha$ such as infliximab and adalimumab (monoclonal antibodies) or etanercept (TNF receptor fusion protein) may be used to limit exacerbated pathology and improve antibiotic activity. These substances are restricted for use in combination with antibiotics (adjuvant treatment) since TNF$\alpha$ is essential for protective immunity and granuloma integrity. Monotherapy with infliximab and other anti-TNF antibodies led to reactivation of latent TB [90].

However, when combined with anti-TB drugs, TNF neutralization enhanced *M. tuberculosis* clearance and reduced lung pathology [91]. A clinical study performed with adjuvant etanercept in patients with pulmonary TB and HIV showed a trend towards improved outcome when the TNF blocker was added to the antibiotic regimen [92].

TNF signaling is also the main target of other HDT candidates such as thalidomide, phosphodiesterase inhibitors or Janus kinase (JAK) inhibitors. Thalidomide has potent anti-inflammatory properties which led to successful application of the drug in cases where anti-TB or HIV treatment triggered hyperinflammatory syndromes such as paradoxical reactions or immune reconstitution syndrome (IRIS). A general application for adjuvant treatment approaches may be hampered due to side effects as seen in a study performed with children suffering from TB meningitis [93]. Phosphodiesterase (PDE) inhibitors seem to be more promising for broad application in TB patients. PDEs degrade cyclic AMP (cAMP), a second messenger negatively regulating TNF levels. Decreased levels of cAMP stimulate TNFα secretion, thus making PDE inhibitors interesting HDT targets. Among the five PDE subtypes, targeting PDE4 seems to be the most promising option in TB with adjuvant use of inhibitors leading to improved outcome in several animal models [94,95]. A phase II clinical trial with the PDE4 inhibitor CC-11050 is ongoing (Table 1).

### 2.4.5. Targeting Matrix Metalloproteinases for Improved Tissue Repair

Imbalanced inflammation eventually results in host tissue destruction, cavitation and dissemination of bacteria. A main driver of these end-stage events are matrix metalloproteinases (MMP) [96]. Once released from activated or necrotic cells, these zinc dependent proteases cleave the extracellular matrix, mostly collagen, and inhibition with small molecules should restrict tissue damage and exacerbation of the disease. Several in vitro and ex vivo studies identified elevated MMP levels in *M. tuberculosis* infected cells or tissue indicating that these enzymes are engaged [96]. In particular, Andrade and colleagues [97] evaluated the interplay between the levels of MMP and heme oxygenase-1 (HO) and discovered that the abundance of these two markers in plasma correlates with different inflammatory profiles and clinical presentations of TB. To date, the only FDA approved MMP inhibitor is doxycycline, an antibiotic with a dual mechanism of action targeting primarily MMP1 and MMP9. The drug suppressed MMP1 and 9 activities in *M. tuberculosis* infected primary human macrophages [98]. In the same study, doxycycline treatment of *M. tuberculosis*-infected guinea pigs led to reduction of the lung bacterial load compared to untreated animals. However, it is important to note that the substance shows a significant growth inhibitory effect on *M. tuberculosis* in broth (MIC 2.5 µg/mL) making it difficult to differentiate between selective host directed and antibacterial effects in these experiments. Experiments with more selective MMP inhibitors such as marimastat (BB-2516), a collagen peptidomimetic broad spectrum MMP inhibitor, showed adjuvant activity in *M. tuberculosis*-infected mice when combined with isoniazid or rifampicin [99]. In contrast to doxycycline, monotherapy with marimastat had no effect on lung bacterial burden [99].

**Table 1.** Candidate compounds for host-directed therapy (HDT) against tuberculosis (TB).

| HDT Effect | Compound | Target or Mode of Action | Notes | Clinical Trials (ClinicalTrials.gov) | References |
|---|---|---|---|---|---|
| Promote phagosome maturation and enhance autophagy | Rapamycin (sirolimus) | Inhibition of mTOR | Metabolized by CYP3A4 | – | [11–14] |
| | Everolimus | Inhibition of mTOR, rapamycin analogue | Anti-cancer agent | NCT02968927 | [18,22,23] |
| | Metformin | Activates AMPK | Used to treat diabetes | Phase 2 studies planned CTRI/2018/01/011176 | [25] |
| | Imatinib | Inhibition of BCR-ABL tyrosine kinase | Used to treat leukaemia | NCT03891901 | [35,36] |
| | Statins | Inhibition of cholesterol biosynthetic pathway | Cholesterol is relevant in *M. tuberculosis* persistence | NCT03882177 NCT03456102 NCT04147286 | [37,40] |
| Induce anti-microbial peptides | Vitamin D | Promotes synthesis of cathelicidin LL37 | Variable results in clinical trials | NCT00918086 NCT01722396 NCT01130311 NCT01244204 NCT00677339 NCT01698476 NCT01137370 (all completed) | [44,47–50] |
| | Vitamin A | Promotes acidification of phagosome | Inconsistent results in rats and humans | NCT00057434 (completed) | [52–56] |
| | 4-phenylbutyrate | Inhibition of histone deacetylase | Promotes autophagy | NCT01580007 NCT01698476 (all completed) | [57–59] |
| | Beta defensin 2 (HBD-2) | Reduces *M. tuberculosis* multiplication | High costs and poor stability | – | [60–62] |

**Table 1.** *Cont.*

| HDT Effect | Compound | Target or Mode of Action | Notes | Clinical Trials (ClinicalTrials.gov) | References |
|---|---|---|---|---|---|
| Adjunct cytokine therapy | Interferon gamma (IFN-γ) | Promotes autophagy and phagosome maturation | Reduces time to sputum conversion | NCT00201123 NCT00001407 (all completed) | [63–66] |
| | Interleukin 2 (IL-2) | Enhances cell-mediated response to infection | Contrasting results in clinical trials | NCT03069534 | [63,68] |
| Reduce inflammation/Inhibit necrotic cell death | Corticosteroids | Multiple anti-inflammatory effects | Standard of care for CNS TB. Other forms of TB may require high doses for beneficial effects leading to unwanted side effects | Multiple clinical trials. See meta-analysis in Critchley et al. 2013 and 2014 | [69–72] |
| | P38 MAPK inhibitors | Protect cells from mitochondria-induced necrosis | | | [77] |
| | Ferrostatin | Decrease of glutathione peroxidase-4 (Gpx4) levels | Mouse study showing beneficial effect | | [86] |
| | Infliximab, adalimumab, etanercept | Inhibition of TNFα | Restricted for use in combination with antibiotics | | [90–92] |
| | CC-11050 | Phosphodiesterase (PDE) inhibition | | NCT02968927 | [94,95] |
| | Doxycycline, marimastat (BB-2516) | Inhibition of matrix metalloproteinases | Doxycycline shows growth inhibition of *M. tuberculosis*, effects probably not purely host directed | NCT02774993 | [99] |
| | NSAID: aspirin, ibuprofen, diclofenac, etoricoxib, indomethacin | Cyclooxygenase 1 and/or 2 inhibition | | NCT02781909 NCT02602509 NCT02503839 | [78] |

## 3. Targeting Bacterial Virulence

### 3.1. The ESX-1 Secretion System

Lately, the interest in finding novel lead compounds, which prevent infection and dissemination by inhibiting bacterial virulence factors, has increased. These anti-virulence molecules target one or more proteins in the virulence machinery with one prominent example being the ESX-1 secretion system as ESX-1 deletion mutants show strongly attenuated phenotypes in vitro and in vivo [100]. ESX-1 is a type VII secretion system essential for host cell infection, bacterial spread and macrophage escape but not for bacterial growth in axenic cultures [101]. In a whole-cell-based phenotypic screening assay selecting for compounds that abrogate ESX-1 dependent host cell death, the two ESX-1 inhibitors BTP15 and BBH7 have been found and characterized [102]. BTP15 inhibits the histidine kinase MprB that regulates ESX-1 via the *espA-espC-espD* operon. BBH7 on the other hand disturbs metal-ion homeostasis leading to zinc stress and thus hindering secretion of ESX-1 substrates such as EsxA and EsxB (Figure 1). These inhibitors can also be used to abrogate ESX-1 dependent activation of the cytosolic DNA sensor cyclic GMP-AMP synthase (cGAS), a main driver of type I interferon (IFN) secretion, thus nicely linking anti-virulence drugs to modulation of the inflammatory response [103].

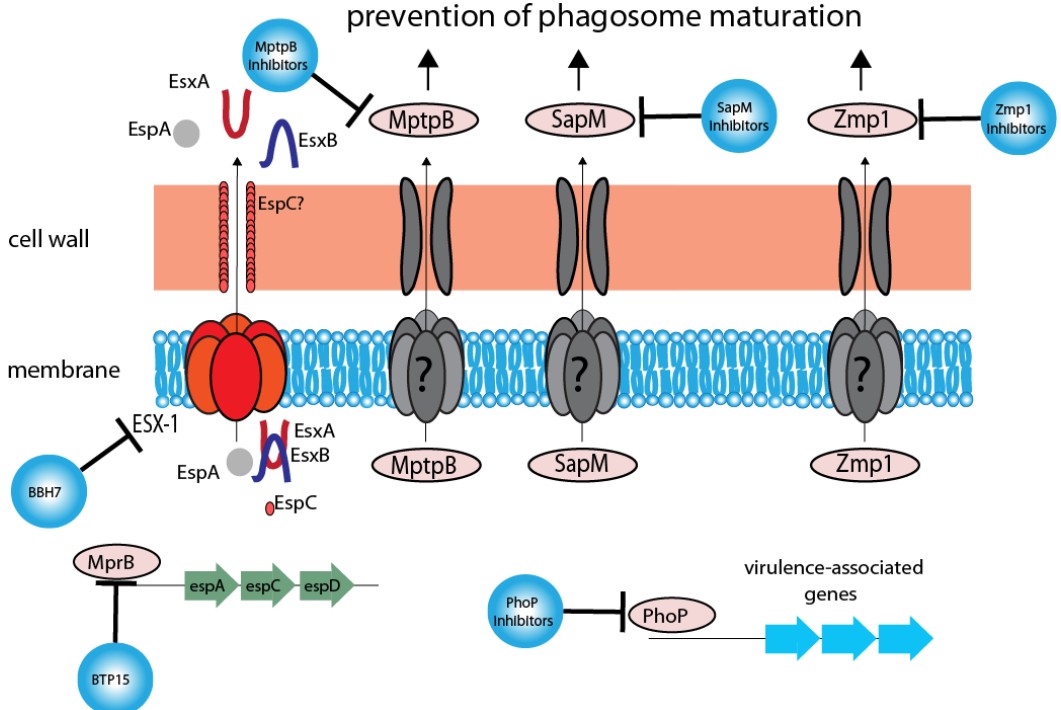

**Figure 1.** Schematic overview of antivirulence targets in *M. tuberculosis*.

### 3.2. PhoPR Inhibitors

The PhoPR two-component system plays a central role in regulating the expression of several proteins relevant for virulence of *M. tuberculosis* as mutants deficient in the effector response regulator PhoP show attenuated growth in infected THP-1 cells and in mice [104]. A microarray-based transcriptional profiling study of *M. tuberculosis* strain H37Rv revealed 110 genes that have been differently expressed in PhoP-deficient mutants [105]. This attenuated strain harbors a single nucleotide polymorphism (S219L) in the DNA-binding domain of PhoP resulting in a reduced DNA-binding capacity [105]. Further studies revealed that PhoP is involved in regulating ESX-1 and in biosynthesis of cell wall components such as sulfolipids, polyacyltrehaloses and diacyltrehaloses [106–108]. Two different approaches identified inhibitors of the PhoPR regulon.

### 3.2.1. Ethoxzolamide

Using a pH-inducible fluorescence reporter system Johnson et al. phenotypically screened for inhibitors of the PhoPR regulon (Figure 1). This screening discovered the carbonic anhydrase inhibitor ethoxzolamide, which inhibits PhoPR while not reducing mycobacterial growth in vitro. Chromatin immunoprecipitation followed by deep sequencing (ChIP-seq) of compound-exposed *M. tuberculosis* cultivated in medium at pH 5.7 showed downregulation of several PhoPR regulated genes involved in lipid synthesis, carbon metabolism and virulence. In addition, the presence of ethoxzolamide did not modulate the expression levels of PhoP itself indicating that the substance acts as a direct inhibitor of the core PhoPR regulon [109].

### 3.2.2. Inhibitors of the PhoP-DNA Complex

Three active compounds (NCGC00093547, NCGC00244580 and NCGC00161636) that directly bind to PhoP and therefore inhibit PhoP-DNA interactions were found in a screening assay based on Foster resonance energy transfer (FRET). For this screening a DNA-Protein complex of Cy3-labeled DNA and Cy5-labeled PhoP protein was exposed to compounds of interest. Inhibitors of this DNA-Protein complex led to dissociation and consequently to a reduced FRET signal [110]. Direct binding of inhibitors to PhoP was confirmed by thermal shift assays in which target-bound inhibitors stabilize the protein and increase the melting temperature, which can be quantified using the fluorescence signal of fluorophore-protein complexes. Compounds NCGC00093547 and NCGC00161636 increased PhoP melting temperature by 14 °C and 18 °C with an $IC_{50}$ of 15.6 and 15.5 µM, respectively. Data on in vivo or ex vivo activity of these compounds is not available yet.

### 3.3. Phagosomal Regulation/Hindering Intracellular Survival

### 3.3.1. MptpB Inhibitors

The *M. tuberculosis* protein-tyrosine-phosphatase B (MptpB) is another putative target for anti-virulence compounds (Figure 1). This kinase is secreted into the cytoplasm of host macrophages allowing for inhibition outside the thick and difficult to overcome mycobacterial cell wall [111]. The function of MptpB is not fully described yet, but the protein has been reported to be necessary for bacterial survival in guinea pigs [112]. So far, it was shown that MptpB dephosphorylates host phosphotyrosine substrates, phosphoserine/threonine substrates and phosphoinositides, with the latter being essential for host macrophage maturation [113].

Several isoxazole-based molecules were created to block the primary and secondary phosphate-binding pockets of MptpB followed by phenotypic testing for activity. In these ex vivo assays, the compounds led to a reduction of mycobacterial burden in macrophages (J774 and THP-1) and in a guinea pig model, without affecting extracellular growth in broth. Additionally, attenuated growth of MDR strains of *M. tuberculosis* in the presence of compound 13 was shown in macrophages. In addition, this inhibitor caused increased sensitivity of a BCG strain to rifampicin and isoniazid in an ex vivo macrophage infection experiment [114]. A similar effect has not been published for *M. tuberculosis* yet.

### 3.3.2. SapM Inhibitors

Another virulence factor that affects phagocytosis and phagosome formation is the secreted acid phosphatase M (SapM) (Figure 1). SapM shows activity as a monoester alkaline phosphatase and targets two phosphoinositides ($PI(4,5)P_2$ and PI3P) important for phagosome maturation [115]. In inhibition studies, it was shown that 2-phospho-L-ascorbic acid interferes with SapM activity without attenuating extracellular bacterial growth. At 4 mM, this drug could reduce intracellular growth of *M. tuberculosis* by 39% [116].

### 3.3.3. Zmp1 Inhibitors

Although its role is not fully understood yet, Zmp1 is involved in mycobacterial pathogenicity as it inhibits the inflammasome and therefore prevents phagosome maturation (Figure 1) [117]. Zinc peptidases like Zmp1 are often inhibited by molecules with specific zinc binding groups (ZBG) like 8-hydroxyquinolines. 8-hydroxyquinolines are already in use as metal-interacting structures in pharmacological applications [118]. Based on this, Vickers et al. synthesized compounds consisting of an 8-hydroxyquinoline ring and a hydroxamate moiety and isosteric analogues of these. One 8-hydroxyquinoline-2-hydroxamate derivative showed a reduction in colony forming units (CFU) in infected J774 mouse macrophages while no extracellular, anti-mycobacterial activity was observed. Treatment of infected human monocyte-derived macrophages with this substance led to a decrease in bacterial burden in a dose-dependent matter. In in-vitro inhibition assays the compound inhibited Zmp1 with an $IC_{50}$ of 0.011 μM [119].

### 3.4. Stress Associated Approaches

#### DosRST Signaling

*M. tuberculosis* exploits its two-component system DosRST to establish a dormant state of nonreplicating persistence (NRP) [120]. As *dosRST* mutants show attenuated growth in animal models, including nonhuman primates and guinea pigs, DosRST might be a potential target to reduce mycobacterial virulence [121]. When investigating compounds for their effect on the DosRST system, Zheng et al. identified two candidates (HC104A and HC106A) which interact with distinct members of the two-component system and decrease production of hypoxia-induced triacylglycerol by around 50% during NRP [122]. In a hypoxic shift-down model using a DosR-dependent fluorescent strain CDC1551(*hspX'::*GFP) HC106A was found to affect *M. tuberculosis* survival during NRP. HC104A on the other hand did not attenuate growth in this setting. The in vivo relevance for these interesting findings still needs to be established in a suitable animal model.

## 4. Conclusions

Several HDT approaches are currently being tested in a number of preclinical and clinical trials as adjuvants complementing conventional anti-TB treatment. Clinical trial results provided in the near future will present an important milestone for the implementation of HDT in routine clinical use. While representing a promising therapeutic approach in theory, most compounds targeting mycobacterial virulence factors lack in vivo proof of principle data.

**Author Contributions:** All of the authors participated in writing and reviewing the manuscript. All authors have read and agreed to the published version of the manuscript.

**Funding:** J.R. receives funding from the Thematic Translational Unit Tuberculosis (TTU TB, grant number TTU 02.806 and 02.905) of the German Centre of Infection Research (DZIF), the German Research Foundation (DFG RY 159, CRC1403), the Centre for Molecular Medicine Cologne (ZMMK–CAP8). J.R. has received funding from the Innovative Medicines Initiative 2 Joint Undertaking (JU) under grant agreement No 853989. The JU receives support from the European Union's Horizon 2020 research and innovation programme and EFPIA and Global Alliance for TB Drug Development Non Profit Organisation, Bill & Melinda Gates Foundation, University Of Dundee.

**Conflicts of Interest:** The authors declare no conflicts of interest.

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
