# Peer review of "Host-Directed Therapies and Anti-Virulence Compounds to Address Anti-Microbial Resistant Tuberculosis Infection"

_applsci, doi:10.3390/app10082688_

Round 1

Reviewer 1 Report

This is a  good comprehensive review of the different approaches for Host-directed therapies (HDT) and anti anti-virulence strategies to address resistant TB. To fully understand the present knowledge and possibilities for future treatment, some additional discussion would be valuable:

  1. The discussion of necrosis of neutrophils and release of virulent Mtb (ref 74), could be expanded to include efferocytosis of apoptotic cells which may in fact strengthen the function of macrophages.
  2. The role of ROS and NO should be discussed. Both these compounds have are important both for their anti-microbial activity and their regulatory role. It has been shown that both ROS and NO inhibit neutrophil recruitment to the granuloma, and that inhibition could aggravate the inflammation. Furthermore, enhancing NO production with arginine can facilitate TB outcome.
  3. The role of inflammasomes  and IL1-ß should be discussed, since they may play a role in IRIS, and extrapulmonary TB.
  4. Minor point: page 2:78; "found to improve the severity of TB disease in humans" should reduce the severity.

Author Response

Reviewer 1

This is a good comprehensive review of the different approaches for Host-directed therapies (HDT) and anti-virulence strategies to address resistant TB. To fully understand the present knowledge and possibilities for future treatment, some additional discussion would be valuable:

  1. The discussion of necrosis of neutrophils and release of virulent Mtb (ref 74), could be expanded to include efferocytosis of apoptotic cells which may in fact strengthen the function of macrophages.

Response: as suggested by the Reviewer, we expanded the section dedicated to necrosis and neutrophils and included efferocytosis. The new paragraph (lines 251-255) reads as follows: “In addition to ferroptosis, efferocytosis (the physiological process of removing apoptotic cells by macrophages) is an anti-bacterial mechanism that seems to play a relevant role in TB as well. Indeed, efferocytosis of apoptotic neutrophils was shown to improve control of M. tuberculosis in an in vitro model of HIV-M. tuberculosis macrophage co-infection”.

  1. The role of ROS and NO should be discussed. Both these compounds have are important both for their anti-microbial activity and their regulatory role. It has been shown that both ROS and NO inhibit neutrophil recruitment to the granuloma, and that inhibition could aggravate the inflammation. Furthermore, enhancing NO production with arginine can facilitate TB outcome.

Response: thank you for this useful comment. We have now included a section on the activity of ROS and NO in neutrophil recruitment (lines 240-245), which reads: “In addition, ROS and nitric oxide (NO) have been found to show antimicrobial activity and to modulate neutrophil recruitment to the granuloma. While ROS seems to increase cytokine production and to inhibit inflammasome activation, NO shows a regulatory effect on macrophages with increased expression of HIF-1α (hypoxia-inducible factor 1 alpha) and repression of NF-κB (nuclear factor kappa-light-chain-enhancer of activated B cells).”

  1. The role of inflammasomes and IL1-ß should be discussed, since they may play a role in IRIS, and extrapulmonary TB.

Response: the role of inflammasome and IL-1b have been discussed in the revised version of the manuscript. Please see lines 198-202: “Investigations at the molecular level proved that dysregulation of inflammasome signalling and of secretion of various cytokines, including IL-1b, was associated with TB-IRIS in patients infected by HIV, thus supporting the inclusion of corticosteroids in the treatment of TB patients at risk of developing IRIS”.

  1. Minor point: page 2:78; "found to improve the severity of TB disease in humans" should reduce the severity.

Response: the sentence has been corrected as suggested by the Reviewer. “Improve” has been replaced by “reduce”.

Reviewer 2 Report

Overall, this is a well written overview of HDT in TB. The manuscript is ready for dissemination with minimal changes. 

Some small questions, suggestions and concerns: 

  1. I would not use the term "gold standard" for the use of corticosteroids for TBM. The classic manuscript actually only had a p-value of 0.06. Further, Dr. Ramakrishnan's group has demonstrated that some individuals have detrimental response to steroids, while others benefit. it is unlikely to be a one-sized fits all response. (See Tobin and Roca et al 2012, 2013)
  2. In the ferroptosis section, consider interpreting the manuscript by Andrade and Sher from 2015 in JI. 
  3. Considering including a section on cox and leukotriene inhibitors. (See above references for Tobin, Roca and Ramakrishnan). 
  4. Considering discussing Dan Barber's few studies about IFNg being detrimental. Is this why why steroids are beneficial? 

Author Response

Reviewer 2

Overall, this is a well written overview of HDT in TB. The manuscript is ready for dissemination with minimal changes.

Some small questions, suggestions and concerns:

  1. I would not use the term "gold standard" for the use of corticosteroids for TBM. The classic manuscript actually only had a p-value of 0.06. Further, Dr. Ramakrishnan's group has demonstrated that some individuals have detrimental response to steroids, while others benefit. it is unlikely to be a one-sized fits all response. (See Tobin and Roca et al 2012, 2013)

Response: thank you for pointing this out. The statement has been modified and now reads “Addition of dexamethasone or prednisolone, two potent corticosteroids, to the antibiotic regimen for treatment of TB meningitis improves survival and is considered as a valid therapeutic approach for TB affecting the central nervous system (CNS), although care should be taken since individual responses to steroid treatment might differ” (please see lines 186-190).

  1. In the ferroptosis section, consider interpreting the manuscript by Andrade and Sher from 2015 in JI.

Response: we thank the Reviewer for suggesting the work by Andrade and Sher, which was added to the bibliography and duly cited in the section on metalloproteinases as we deemed it more suitable for this rather than for the ferroptosis paragraph. Please see lines 291-294:In particular, Andrade and colleagues evaluated the interplay between the levels of MMP and heme oxygenase-1 (HO) and discovered that the abundance of these two markers in plasma correlates with different inflammatory profiles and clinical presentations of TB”.

  1. Considering including a section on cox and leukotriene inhibitors. (See above references for Tobin, Roca and Ramakrishnan).

Response: Cyclooxygenase (Cox) inhibitors were mentioned in the submitted version of the manuscript, in the section dedicated to “Non-steroidal anti-inflammatory drugs (NSAID)” and in Table 1. We added two sentences about leukotriene inhibitors in the same paragraph, according to the Reviewer’s advice. The paragraph reads as follows (lines 226-231): “Another category of anti-inflammatory drugs is represented by leukotriene receptor antagonists, such as zafirlukast, which was reported to have anti-mycobacterial activity in vitro and cause alterations in the transcription profile in M. tuberculosis. The potential of these drugs in HDT against TB deserves deeper investigation given the role for leukotriene A(4) hydrolase (LTA4H) demonstrated by Tobin and colleagues in animal models of infection”. The papers by Tobin, Roca and Ramakrishnan have been included in the bibliography.

  1. Considering discussing Dan Barber's few studies about IFNg being detrimental. Is this why steroids are beneficial?

Response: this point has been considered and addressed in the revised version in lines 174-176: “However, the role of IFN-g in controlling TB infection is still under debate, as reported in a study by Sakai and co-workers, who showed that contribution of CD4-T cell derived IFN-g is limited and, even worse, sometimes detrimental”.

Reviewer 3 Report

General Comments:

  1. Well-written review of approaches to provide new options for tuberculosis treatment.
  2. Excellent and useful Tables

Minor Comments

Lines 111-119. It might help to provide background by including several sentences on vitamin D-deficiency and susceptibility to Mycobacterium tuberculosis.

Line 199. Size of type and italics

Author Response

Reviewer 3

General Comments:

  1. Well-written review of approaches to provide new options for tuberculosis treatment.
  2. Excellent and useful Tables

Response: we thank the Reviewer for the appreciation.

Minor Comments

Lines 111-119. It might help to provide background by including several sentences on vitamin D-deficiency and susceptibility to Mycobacterium tuberculosis.

Response: the potential association between vitamin D and TB was partially described in the submitted version of the manuscript. However, we have now provided an additional comment and cited a recent study conducted in Lima, Peru. Please see lines 143-147: “A recently published study by Aibana and co-workers suggested that vitamin D deficiency is associated with increased probability of developing TB in HIV-positive people. However, further investigations are needed to clarify whether vitamin D supplementation might play a significant role in reducing the risk of TB”.

Line 199. Size of type and italics

Response: we checked the spelling throughout the whole manuscript and corrected as appropriate.

In the end, we would like to thank the Reviewers for their insightful comments and constructive criticism, which helped us improve the manuscript. We hope that it will be now found acceptable for publication and thank you for your assistance in this matter.